# Easy, Fast Self-Heating Polyurethane Nanocomposite with the Introduction of Thermally Annealed Carbon Nanotubes Using Near-Infrared Lased Irradiation

**DOI:** 10.3390/ma15238463

**Published:** 2022-11-28

**Authors:** Hyunsung Jeong, Sooyeon Ryu, Young Nam Kim, Yu-Mi Ha, Chetna Tewari, Seong Yun Kim, Jung Kyu Kim, Yong Chae Jung

**Affiliations:** 1Institute of Advanced Composite Materials, Korea Institute of Science and Technology (KIST), 92 Chudong-ro, Bongdong-eup, Wanju-gun 55324, Republic of Korea; 2School of Chemical Engineering Building, Sungkyunkwan University, II2066 Seobu-ro, Jangan-gu, Suwon 16419, Republic of Korea; 3Carbon Materials and Engineering, Jeonbuk National University, Baekje-daero, Deokjin-gu, Jeonju-si 54896, Republic of Korea; 4Advanced Materials Division, Korea Research Institute of Chemical Technology (KRICT), 141 Gajeong-Ro, Yuseong-gu, Daejeon 34114, Republic of Korea

**Keywords:** Polyurethane (PU), thermal annealed, carbon nanotube, photothermal effect, Near-infrared radiation (NIR)

## Abstract

In this study, high-crystallinity single walled carbon nanotubes (H-SWNTs) were prepared by high-temperature thermal annealing at 1800 °C and a self-heating shape memory polyurethane nanocomposite with excellent self-heating characteristics was developed within a few seconds by irradiation with near-infrared rays. With a simple method (heat treatment), impurities at the surface of H-SWNTs were removed and at the same time the amorphous structure converted into a crystalline structure, improving crystallinity. Therefore, high conductivity (electric, thermal) and interfacial affinity with PU were increased, resulting in improved mechanical, thermal and electric properties. The electrical conductivity of neat polyurethane was enhanced from ~10^–11^ S/cm to 4.72 × 10^−8^ S/cm, 1.07 × 10^−6^ and 4.66 × 10^−6^ S/cm, while the thermal conductivity was enhanced up to 60% from 0.21 W/mK, 0.265 W/mK and 0.338 W/mK for the composites of 1, 3 and 5 wt%, respectively. Further, to achieve an effective photothermal effect, H-SWNTs were selected as nanofillers to reduce energy loss while increasing light-absorption efficiency. Thereafter, near-infrared rays of 818 nm were directly irradiated onto the nanocomposite film to induce photothermal properties arising from the local surface plasmon resonance effect on the CNT surface. A self-heating shape memory composite material that rapidly heated to 270 °C within 1 min was developed, even when only 3 wt.% of H-SWNTs were added. The results of this study can be used to guide the development of heat-generating coating materials and de-icing materials for the wing and body structures of automobiles or airplanes, depending on the molding method.

## 1. Introduction

Polyurethane (PU) is a representative engineering plastic with excellent mechanical properties, such as good heat and wear resistance, a high elongation rate, and excellent workability. It has been widely applied in adhesives, coatings, rubber, and biomaterials with good biocompatibility, high chemical resistance, and excellent processability [1,2,3,4,5]. In addition to the excellent physical properties of such materials, the addition of a carbon-based nanofiller to the nanocomposite material can induce new behaviors such as thermal conductivity and electrical conductivity [6,7,8,9,10].

There are many applications of carbon nanotube (CNT)-based nanocomposites, owing to their high aspect ratio, good charge transport properties, and high electrical and thermal properties [11,12]. In particular, single-walled carbon nanotubes (SWCNTs) have the potential to be used in electronic devices, owing to their excellent electrical and mechanical properties [13,14,15,16]. However, owing to factors such as van der Waals forces, it is extremely difficult to align and disperse CNTs in a polymer matrix. The biggest challenge in translating the excellent properties of CNTs into composite structures is the efficient dispersion of CNTs in a polymer matrix, assessment of the dispersion, and alignment and control of the CNTs in the matrix [17,18].

Various surface treatment methods have been developed to alter the thermal and electrical properties of CNTs. For example, high-temperature heat treatment and doping of hetero-elements can allow the control of CNT transport characteristics, which is effective for improving their conductivity by controlling their energy bandgaps; this method can be applied to develop new nanocomposites [19,20,21].

We previously reported a photothermal effect using near-infrared radiation (NIR) after preparing a PU nanocomposite using boron-doped CNTs [6]. However, the chemical doping method is disadvantageous in that the physical properties of SWCNTs tend to decrease continuously with changes in their external environment, such as increased temperature and humidity [22,23]. Recent studies have addressed this issue.

The photothermal effect is defined as a physical phenomenon in which, when a material is irradiated with light energy, the irradiated light energy is absorbed and then released as heat according to the properties and characteristics of the material. To use the photothermal effect in a self-heating shape memory composite material, the selection of nanofillers that increase the light absorption efficiency while decreasing the fluorescence quantum yield and controlling the interfacial properties with the polymer matrix should be considered together.

CNTs were selected as representative candidates. CNTs can convert absorbed light into heat with strong absorbance over a wider wavelength range than that of metal nanoparticles. The photothermal effect of CNTs depends on their dispersion, type, and morphology on the substrate. These properties have been applied in various fields such as nano-thermal derivatives, photothermal therapy, imaging, actuators, and sensors [24,25,26,27,28,29,30,31].

In this study, SWCNT nanofillers annealed at high temperatures were combined with PU to produce nanocomposites. In order to overcome the disadvantages of CNT modification through doping method, highly crystalline CNTs were prepared through high-temperature heat treatment. The method through high heat treatment removes impurities on the surface and has high crystallinity through crystallization of an amorphous material, so that CNTs having excellent electrical and mechanical properties can be obtained. The photothermal properties of the nanocomposites were evaluated by irradiating the prepared PU nanocomposite films with NIR light at 808 nm. The characteristics of the filler, content, and degree of dispersion, as well as the mechanical, thermal, and electrical properties of the nanocomposite material were evaluated.

## 2. Materials and Methods

### 2.1. Preparation of Highly Thermally Annealed SWNTs (H-SWNTs)

Single-walled nanotubes (Product No. SA230, Nanosolution Co., Ltd., Uiwang, Republic of Korea) with an ideal diameter (~1.4–1.7 nm), length (5–20 μm), tap density (0.02 g/cc) and high purity (>95 wt.%) were synthesized by arc discharge process. The highly thermally annealed SWCNTs (H-SWNTs) were obtained by annealing at 1800 °C for 30 min in a graphite furnace operating in an argon atmosphere. The H-SWNTs were then kept in a desiccator for future use.

### 2.2. Synthesis of Shape Memory PU Block Copolymer and Preparation of Nanocomposite

Poly(tetramethylene glycol) (PTMG, MW = 1800 g/mol, 1 mmol), 4,4′-methylene bis(phenylisocyanate) (MDI, Junsei Chemical, 4 mmol), and 1,4-butandiol (BD, Ducksan Chemical, 3 mmol) were used for the synthesis of the shape memory PU block copolymer (SHPU) [17]. In a 500 mL four-necked cylindrical vessel equipped with a mechanical stirrer, calculated amounts of PTMG and MDI in 100 mL of freshly distilled dimethylacetamide were stirred under nitrogen at 80 °C for 3 h to form the prepolymer. Subsequently, the reaction was controlled by cooling at 30 °C for 4 h. In the second step, BD was added dropwise to the reaction mixture depending on the MDI/PTMG ratio. When polymerization was complete, the PU was removed using a solvent under vacuum, and it was further solidified by storing it in an oven at 100 °C for 12 h. The synthesized SHPUs were dissolved uniformly in N-methyl-2-pyrrolidone (NMP) at a concentration of 10 wt.% and stirred constantly to ensure homogeneity.

The homogeneously dispersed H-SWNT suspension was prepared by adding nanotubes to NMP and then sonicating for 1 h (horn-type SONICS VCX, 750 W, 20 kHz, amplitude of 80%). The synthesized SHPU solution (re-dissolved in NMP) was poured into the nanotube suspension and mixed using a planetary centrifugal mixer (THINKY ARM-310 Centrifugal Mixer). Finally, we prepared a composite film by casting the H-SWNTs/SHPU suspension on chalets and drying at 50 °C for 3 h and at 110 °C for 12 h in vacuum to remove the residual solvent. The resulting samples were obtained as films with thicknesses of 100–120 µm. SHPU nanocomposites with H-SWNT loadings of 1, 3, and 5 wt.% were fabricated, and their specific compositions are summarized in Table 1.

### 2.3. Characterizations

Raman spectroscopy (Renishaw inVia-reflex Raman microscope, 514 nm laser line) was used to monitor the changes in the optical properties of the CNTs before and after high-temperature thermal annealing. Wide-angle X-ray diffraction (WAXD, Thermo Scientific, Waltham, MA, USA) analysis was performed to identify the structures of neat PU, P-SW@PU, H-HW@PU composites. X-ray photoelectron spectroscopy (XPS, K-Alpha, ThermoFisher Scientific, Waltham, MA, USA) was performed using a Mg Kα X-ray source with a 10-mA emission current and a 10 kV accelerating voltage. The thermal stabilities of the nanocomposites were examined using thermogravimetric analysis (TGA) (Q50, TA Instruments, New Castle, DE, USA) in the temperature range from room temperature to 900 °C at a rate of 10 °C/min in nitrogen.

Fourier-transform infrared (FT-IR) spectroscopic measurements were performed to identify the structure of the SHPU and H-SWNTs using an IR Prestige-21 (Shimadzu) equipped with an attenuated total reflectance accessory. The cross-sectional morphology of the SHPU nanocomposites was examined by field-emission scanning electron microscopy (FE-SEM, NOVA nano SEM 450, FEI, Hillsboro, AZ, USA) using Pt-coated samples. The mechanical properties were evaluated at room temperature according to the ASTM D638 test method using a universal testing machine (UTM 5567A, Instron, Norwood, MA, USA) with dog-bone-type dumbbell specimens. The dimensions of the specimens were 30 (length) × 5 (width) × 5 (narrow portion length) × 2 (narrow portion width) × 0.20 (thickness) mm. The measurement conditions were as follows: 10 mm gauge length; 20 mm/min crosshead speed; and 100 N load cell. At least five samples were tested, and the average was used. An NIR laser was used to measure the heat-generation characteristics of the nanocomposite materials. Photothermal measurements of the pristine SWNTs and the H-SWNT-incorporated PU nanocomposite were performed using the NIR laser (808 nm, CNI laser) and the laser power density of 1.5 W/cm^2^. The NIR laser spot size of 0.8–0.5 cm^2^ was used.

## 3. Results

Figure 1 depicts the experimental method used in this study and expresses the synthesis of PU and the manufacture of self-heating composites using nanofillers as additives. Figure 1a shows the Raman spectra of the pristine SWNTs (P-SWNTs) and H-SWNTs obtained at a wavelength of 514 nm. The crystal structures of the CNTs before and after high-temperature heat treatment were also analyzed. All samples showed typical SWCNT Raman spectroscopic characteristics. In particular, when checking the radial breathing mode (RBM) peak at 100–350 cm^−1^, it can be seen that the tube diameter was approximately 1.25 nm (deVRBM = 248/d). However, the change in the diameter of the tube owing to heat treatment could not be confirmed. The G-band at 1591 cm^−1^ and the D-band at 1347 cm^−1^ resulting from the E2g mode were observed [32,33], and the heat-treated H-SWNTs complemented the structural bonding caused by the removal of amorphous carbon from the tube surface. This can be attributed to the two-dimensional growth of the graphene layer along the longitudinal direction of the CNT [34]. Therefore, the intensity of the D-band decreases. Additionally, by calculating the R value, i.e., I_D_/I_G_ (0.003 for H-SWNTs and 0.02 for P-SWNTs), it is clear that the graphite structure and crystalline structure of the heat-treated SWCNT (H-SWNTs) were more developed than those of the P-SWNTs [35,36]. To observe the change in the SWCNT crystallinity according to the structural changes via Raman spectroscopy, the interlayer distance (d002) of the tube was calculated before and after the high-temperature heat treatment through WAXD analysis. The interplanar distance (d-spacing) of carbon nanotubes is obtained from the (002) peak of the XRD profile following Bragg’s law:nλ = 2d∙sinθ
where n is an integer, λ is the X-ray wavelength, d is the interplanar distance, and θ is the diffraction angle.

As shown in Figure 1b, the interplanar distance was narrowed by 0.0075 Å from 3.3632 Å to 3.3557 Å after heat treatment. Moreover, in the 2θ range of 40–60°, both samples show (100), (102), and (004) peaks, corresponding to the hexagonal ring structure of the graphite sheet that forms the CNT. These peaks also show the heat treatment effect, confirming the development of in-plane ordering [37].

In addition, Figure 1c shows the TGA data obtained under a nitrogen atmosphere, characterizing the thermal decomposition behavior of the SWNTs during heat treatment. It can be seen that the thermal decomposition of H-SWNTs, whose crystallinity was increased by the removal of amorphous materials from the CNT surface during heat treatment, began at or above approximately 800 °C; 97% of CNTs remained at around 900 °C. However, for the neat SWNT, a weight reduction rate of 80% was observed at the same temperature, and the thermal decomposition temperature was approximately 300 °C lower, in the range of 500–700 °C. That is, high-temperature heat treatment of CNTs increases their thermal stability [38,39,40].

Figure 1d shows the chemical bonding state between the atoms on the CNT surfaces after heat treatment as determined by XPS. Figure 1d and Appendix A, and Appendix A compare the C 1s spectral profiles of the SWNTs before and after the heat treatment. It can be seen that all carbon–oxygen bonding states such as sp^2^ (284.7 eV), sp^3^ (285.2 eV), and C=O (286.1 eV), which are carbon-to-carbon bonds in graphite, change significantly even after heat treatment due to the amorphous carbon the outer surface of the tube [8]. In addition, O1s shows a similar trend, but C=O (533.5 eV) is relatively more developed. However, the relative comparative value of O1s/C1s decreased from 0.08 to 0.03, and the element content of C1s and O1s increased after heat treatment, as shown in Figure 1a–c.

H-SWNTs with crystallinity developed through high-temperature heat treatment were added to PU to prepare a nanocomposite material, and FT-IR was used to observe changes in the structural properties. In Figure 2, the characteristic NH stretching and C=O stretching peaks attributed to PU were confirmed at 3200 cm^−1^ and 1730 cm^−1^, respectively. In addition, in the case of the nanocomposites (P-SW@PU and H-SW@PU) to which fillers were added before and after heat treatment, the OH functional group at 3500 cm^−1^ decreased gradually, and the asymmetric C–H stretching bond at 2925 cm^−1^ and 2845 cm^−1^ was reduced. However, because the synthesized PU nanocomposite was formed by simple physical mixing with CNTs, no significant changes in the peaks could confirm the reaction between the PU backbone and nanofillers.

Tensile testing was used to evaluate the mechanical properties of the PU nanocomposites containing nanofillers, and the elastic modulus, breaking strength, and elongation at break were monitored as the filler content was increased to 1, 3, and 5 wt.% based on the weight of PU as shown Figure 3a,b.

Figure 3a,b show the measurement results of the stress–strain curves of the neat PU specimens and the SHPU nanocomposite with added H-SWNT. The Young’s modulus, breaking strength, and elongation at break of neat PU were 6.86 MPa, 5.59 MPa, and 220%, respectively, whereas the PU nanocomposite material with H-SWNT filler showed significantly improved properties as the amount of added filler increased. In the case of the 5 wt.% sample, the maximum Young’s modulus and breaking strength were increased by 497% and 50.8%, respectively, compared to those of pure PU, while the elongation at break was decreased by 40.9%. The cause of the increase in the Young’s modulus and breaking strength is the entanglement of the H-SWNTs with the surrounding PU, forming a rigid percolation network with rigid domains in the polymer [8,41,42]. This characteristic is interpreted as an interaction effect between the filler and mechanical properties because the degree of percolation networking also increases as the filler content increases. In addition to the filler content, the dispersibility is an important factor. The aforementioned rigid domains were the effect of the filler. If the filler particles in the matrix are strongly agglomerated or have low dispersibility, load transfer to the CNTs is nonuniform when a load is applied to the matrix. Such an effect not only lowers the mechanical strength but also reduces the electrical and thermal conductivities.

Figure 3c shows the measurements of the electrical and thermal conductivities of the PU nanocomposite evaluated above. It was confirmed that the electrical conductivity (~10^–11^ S/cm) of the neat PU was consistent with the values reported reference [43]. As the H-SWNT content was changed from 1 to 3 and 5 wt.%, the electrical conductivity changed from 4.72 × 10^−8^ to 1.07 × 10^−6^ and 4.66 × 10^−6^ S/cm, respectively. In addition, the thermal conductivity increased by up to 60% from 0.21 to 0.265 and 0.338 W/m·K, respectively. Such a result can be seen as an effect of an increase in the crystallinity of the CNTs, as well as an increase in the filler content.

In Figure 3a–c, the properties were affected not only by the filler content but also by the dispersibility. Therefore, we cut the samples to check the dispersion states of the H-SWNT-added PU nanocomposite materials and then observed the cross-sections using SEM (Figure 3d). Figure 3d shows SEM images of cross-sections of PU nanocomposites with different filler contents. From the SEM result, uniformly dispersed CNTs can be observed, and it can be seen that the distribution effect is the same, even at 5 wt.%.

Figure 4 shows the real-time photothermal properties as measured with a thermal imaging camera after irradiating a sample with an NIR laser (818 nm, power density 0.1 W/cm^2^) to evaluate the self-heating properties of the developed shape memory PU nanocomposite. In the case of PU, it can be seen that no temperature change occurs on the surface regardless of the presence of laser irradiation. However, the H-SWNT-containing PU nanocomposite was rapidly heated to 270 °C within 1 min. In particular, as the filler content increased, the maximum exothermic temperature tended to increase, but considering that the filler content showed a similar trend at 3 and 5 wt.%, the filler content required for maximum exothermic performance was judged to be around 3 wt.%.

## 4. Conclusions

In this study, a self-heating nanocomposite material with excellent mechanical, thermal, and electrical properties was manufactured with polyurethane (PU) and the high-temperature heat treated SWCNTs. The high heat treatment was given at 1800 °C to remove the amorphous material from the surface of CNTs and develop more crystalline SWCNTs. First, SWCNTs through high-temperature heat treatment (H-SWNTs) were able to control the structure of CNTs by removing impurities from the surface of CNTs and changing amorphous materials into crystalline structures. Second, when high crystalline H-SWNTs were incorporated instead of P-SWNTs with PU, it shows higher mechanical properties than P-SWNTs. Subsequently, the characteristics of the nanocomposite materials, which were capable of heating from room temperature to 270 °C within 1 min, were investigated by irradiating with NIR light of 818 nm. As a result, using only heat treatment, it was possible to improve the surface characteristics of the nanofiller and develop a material with excellent heat-generating performance, electrical conductivity and mechanical properties.

In the future, this self-heating nanocomposite could be used in all industries, from home heaters to large-area heaters for e-mobility, functional windows for construction, showcase heaters, and heater modules in the printing industry.

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
