# Peer review of "Easy, Fast Self-Heating Polyurethane Nanocomposite with the Introduction of Thermally Annealed Carbon Nanotubes Using Near-Infrared Lased Irradiation"

_materials, 2022, doi:10.3390/ma15238463_

Round 1
Reviewer 1 Report
The research article represents the Fast Self-heating Polyurethane Nanocomposite with the introduction of Thermally Annealed Carbon. I recommend to accept for publication after minor corrections.
Comments
1. Abstract must be summarizing the results and new findings of the manuscript.
2. Authors have to include the synthesis method of SWNT in experimental section.
3. How much BD is added to synthesize of PU?
4. Tensile strength and modulus decreased in the cass of PU composite with 1wt% of SWNT compared to PU, explain?
5. In conclusion, authors should include some good results.
Author Response
Referee #1
The research article represents the Fast Self-heating Polyurethane Nanocomposite with the introduction of Thermally Annealed Carbon. I recommend to accept for publication after minor corrections.
[Comment 1] Abstract must be summarizing the results and new findings of the manuscript.
Reply: Thank you very much for your valuable comment. As per your suggestion we have modified the abstract and incorporated the results and new findings in the abstract.
The modified abstract is given below:
“In this study, high-crystallinity single walled carbon nanotubes (H-SWNTs) were prepared by high-temperature thermal annealing at 1800 °C and a self-heating shape memory polyurethane nanocomposite with excellent self-heating characteristics was developed within a few seconds by irradiation with near-infrared rays. With a simple method (heat treatment), impurities at the surface of H-SWNTs were removed and at the same time the amorphous structure converted into a crystalline structure, improving crystallinity. Therefore, high conductivity (electric, thermal) and interfacial affinity with PU were increased, resulting in improved mechanical, thermal and electric properties. The electrical conductivity of neat polyurethane was enhanced from ~10–11S/cm to 4.72×10−8S/cm, 1.07×10−6 and 4.66×10−6 S/cm, while the thermal conductivity was enhanced up to 60% from 0.21 W/mK, 0.265 W/mK and 0.338 W/mK for the composites of 1, 3 and 5wt% respectively. Further, to achieve an effective photothermal effect, H-SWNTs were selected as nanofillers to reduce energy loss while increasing light-absorption efficiency. Thereafter, near-infrared rays of 818 nm were directly irradiated onto the nanocomposite film to induce photothermal properties arising from the local surface plasmon resonance effect on the CNT surface. A self-heating shape memory composite material that rapidly heated to 270 °C within 1 min was developed, even when only 3 wt.% of H-SWNTs were added. The results of this study can be used to guide the development of heat-generating coating materials and de-icing materials for the wing and body structures of automobiles or airplanes, depending on the molding method.”
As per your suggestion, the abstract have been modified in the revised manuscript. (Page no. -1, lines 20 to 26)
[Comment 2] Authors have to include the synthesis method of SWNT in experimental section.
Reply: Thank you very much for your valuable comment. As per your suggestion, we have modified the experimental section as “Single-walled nanotubes (Product No. SA230, Nanosolution Co., Ltd., Republic of Korea) with an ideal diameter (~1.4–1.7 nm), length (5 20 mm), tap density (0.02 g/cc) and high purity (>95 wt.%) were synthesized by arc discharge process. The highly thermally annealed SWCNTs (H-SWNTs) were obtained by annealing at 1800℃ for 30 min in a graphite furnace operating in an argon atmosphere. The H-SWNTs were then kept in a desiccator for future use.” (Page number -3, lines-7 to 9).
[Comment 3] How much BD is added to synthesize of PU?
Reply: Thank you very much for this important comment. As per your comment, we added a table (given below) in main manuscript which gives information about the amount of material used in the experiment. (Page number - 5, Table-1).
Table R1. Composition table of polyurethane nanocomposite
|
Samples |
Mol% |
Filler contents |
||
|
MDI |
PTMG |
BD |
||
|
Pure PU |
4.0 |
1.0 |
3.0 |
- |
|
Pristine SWCNTs/PU |
1 |
|||
|
3 |
||||
|
5 |
||||
|
*HTT SWCNTs/PU |
1 |
|||
|
3 |
||||
|
5 |
||||
(*HTT: High temperature heat treatment)
[Comment 4] Tensile strength and modulus decreased in the case of PU composite with 1wt% of SWNT compared to PU, explain?
Reply: Thank you for this comment. As the SWCNTs that have not been heat treated have poor affinity with PU and interfaces, and that the composite containing a small amount of SWCNTs has lower mechanical properties than pristine PU because the filler does not form percolation networking in the polymer.
Reference: Ha, Y.-M.; Lim, D.; Lee, S. Y.; Kim, J.; Kim, J. H.; Kim, Y. A.; Jung, Y. C., Enhanced thermal conductivity and mechanical properties of polyurethane composites with the introduction of thermally annealed carbon nanotubes. Macromol.Res. 2017, 25, (10), 1015-1021.
[Comment 5] In conclusion, authors should include some good results.
Reply: Thank you for your valuable comment. As per your suggestion, we modified the conclusion as “In this study, a self-heating nanocomposite material with excellent mechanical, thermal, and electrical properties was manufactured with polyurethane (PU) and the high-temperature heat treated SWCNTs. The high heat treatment was given at 1800°C to remove the amorphous material from the surface of CNTs and develop more crystalline SWCNTs. First, SWCNTs through high-temperature heat treatment (H-SWNTs) were able to control the structure of CNTs by removing impurities from the surface of CNTs and changing amorphous materials into crystalline structures. Second, when high crystalline H-SWNTs were incorporated instead of P-SWNTs with PU, it shows higher mechanical properties than P-SWNTs. Subsequently, the characteristics of the nanocomposite materials, which were capable of heating from room temperature to 270°C within 1 min, were investigated by irradiating with NIR light of 818 nm. As a result, using only heat treatment, it was possible to improve the surface characteristics of the nanofiller and develop a material with excellent heat-generating performance, electrical conductivity and mechanical properties.
In the future, this self-heating nanocomposite could be used in all industries, from home heaters to large-area heaters for e-mobility, functional windows for construction, showcase heaters, and heater modules in the printing industry.” (Page number - 9, lines- 17 to 20).

Reviewer 2 Report
The manuscript describes a self-heating polyurethane nanocomposite with thermally annealed carbon nanotubes using near-infrared lased irradiation. In general, the paper is well written and details about the work were clearly presented. However, the novelty of this work is below the standard papers published in this journal. Overall, the paper's scientific value would be good enough to be published after the authors' major revision of the manuscript considering the following comments:
1. The main novelty of this work must be clearly mentioned in the introduction as compared to other publications.
2. Why does the annealing of SWCNTs affect final results?
3. To discuss Raman spectroscopy results, peak ratios e.g. ID/IG should be calculated.
4. To compare the effect of annealed and pristine SWCNTs on the final properties of PU, I strongly recommend you prepare PU nanocomposites with P-SWCNT and perform mechanical, thermal and electrical measurements to clarify your results with annealed SWCNTs.
5. There are some errors in the manuscript, please revise them carefully.
Author Response
Referee #2
The manuscript describes a self-heating polyurethane nanocomposite with thermally annealed carbon nanotubes using near-infrared laser irradiation. In general, the paper is well written and details about the work were clearly presented. However, the novelty of this work is below the standard papers published in this journal. Overall, the paper's scientific value would be good enough to be published after the authors' major revision of the manuscript considering the following comments:
[Comment 1] The main novelty of this work must be clearly mentioned in the introduction as compared to other publications.
Reply: Thank you for your valuable comment. In this work, the development of the high crystalline SWCNTs has been reported by high-temperature thermal annealing at 1800 °C, and also incorporation of this developed H-SWCNT in polyurethane matrix to enhance its mechanical, electrical and thermal properties for developing self-heating polyurethane nanocomposites. Also, to increase the photothermal effect, surface modification or structural control of CNTs is required. In our previous study, single-walled carbon nanotubes (SWCNTs) were doped with boron to enhance the photothermal effect. However, the doping method may lose its properties in high temperature or humidity. So, the modification of SWCNTs through heat treatment proposed in this study could improve the thermal and electrical properties by controlling the structure of high crystallization without damaging the CNTs. Also, the developed polyurethane composites with H-SWNTs show excellent self-heating properties than the composites made by commercial SWNTs. According to previously reported data, the nanocomposites of commercially available SWCNT and polyurethane have the maximum self-heating temperature of 164°C, which has been increased up to 270°C by incorporating high crystalline SWCNTs even just in one minute.
However, according to reviewer valuable suggestion, we have highlighted the novelty of the work in the last paragraph of the introduction section as “In this study, SWCNT nanofillers annealed at high temperatures were combined with PU to produce nanocomposites. In order to overcome the disadvantages of CNT modification through doping method, highly crystalline CNTs were prepared through high-temperature heat treatment. This high heat treatment method removes impurities on the surface and provides high crystallinity through crystallization of an amorphous material, so that CNTs having excellent electrical and mechanical properties can be obtained. The photothermal properties of the nanocomposites were evaluated by irradiating the prepared PU nanocomposite films with NIR light at 808 nm. The characteristics of the filler, content, and degree of dispersion, as well as the mechanical, thermal, and electrical properties of the nanocomposite material were evaluated” (Page number -2, lines- 38 to 42)
Reference: Ha, Y.-M.; Kim, Y.-O.; Kim, Y.-N.; Kim, J.; Lee, J.-S.; Cho, J. W.; Endo, M.; Muramatsu, H.; Kim, Y. A.; Jung, Y. C., Rapidly self-heating shape memory polyurethane nanocomposite with boron-doped single-walled carbon nanotubes using near-infrared laser. Compos. B. Eng. 2019, 175, 107065.
[Comment 2] Why does the annealing of SWCNTs affect final results?
Reply: Thank you for your valuable comment. The main aim of this study is the development of self-heating composites using SWCNTs. Photothermal properties are due to electrons located on the surface of a nanomaterial being converted into thermal energy by localized surface plasmon resonance (LSPR) by irradiation of a nanomaterial having π electrons and having a size smaller than the wavelength of the light. CNTs have the ability to convert absorbed light into heat energy and convert it into a strong absorbance over a wider wavelength range than metal nanoparticles can. The photothermal effect of nanotubes depends on the dispersion, type, and morphology of the nanotubes on the substrate. Therefore, in this study, thermal and electrical conductivity of CNTs were maximized by increasing the crystallinity of the structure through crystallization of impurities or amorphous materials on the CNT surface by heat treatment of SWCNTs to increase photothermal properties.
Reference:
- Shuba M, Yuko D, Kuzhir P, Maksimenko S, Chigir G, Pyatlitski A, et al. Localized plasmon resonance in boron-doped multiwalled carbon nanotubes. Physical Review B. 2018;97(20):205427.
- Buchs G, Bagiante S, Steele GA. Identifying signatures of photothermal current in a double-gated semiconducting nanotube. Nature communications. 2014;5:4987.
3, Boldor D, Gerbo NM, Monroe WT, Palmer JH, Li Z, Biris AS. Temperature measurement of carbon nanotubes using infrared thermography. Chemistry of materials. 2008;20(12):4011-6.
- Shen X, Viney C, Wang C, Lu JQ. Greatly enhanced thermal contraction at room temperature by carbon nanotubes. Advanced Functional Materials. 2014;24(1):77-85.
[Comment 3] To discuss Raman spectroscopy results, peak ratios e.g. ID/IG should be calculated.
Reply: Thank you for your valuable suggestion. As per your suggestion, we have modified the Raman spectroscopy results by including the peak area ratio of the D and G bands (ID/IG) in the revised manuscript. The calculated R value (ID/IG) of the H-SWNTs is 0.003 while for P-SWNTs is 0.02. The very low ID/IG ratio of H-SWNTs indicates its highly purity and crystallinity. (Page number -6, lines- 4 to 7)
“Additionally, by calculating the R value i.e. ID/IG (0.003 for H-SWNTs and 0.02 for P-SWNTs), it is clear that the graphite structure and crystalline structure of the heat-treated SWCNT (H-SWNTs) were more developed than those of the P-SWNTs”
Reference:
- K. McGuire, N. Gothard, P. Gai, M. Dresselhaus, G. Sumanasekera, A. Rao, Carbon 2005, 43, 219,
- J. Maultzsch, S. Reich, C. Thomsen, S. Webster, R. Czerw, D. Carroll, S. Vieira, P. Birkett, C. A. Rego, Appl. Phys. Lett. 2002, 81, 2647.
Figure R1. Raman spectra of pristine CNT(P-SWCNT) and heat treated SWCNT(H-SWCNTs)
[Comment 4] To compare the effect of annealed and pristine SWCNTs on the final properties of PU, I strongly recommend you prepare PU nanocomposites with P-SWCNT and perform mechanical, thermal and electrical measurements to clarify your results with annealed SWCNTs.
Reply: Thank you for your valuable suggestion. As per your suggestion, the data on PU composite using p-SW is shown below and added to the supplementary material.
Figure R2. Stress-strain curve of pristine SWCNT(P-SW)/PU composites
Figure R3. Compare of conductivity PU composites
Figure R4. Figure S4. Surface temperature p-SW/PU composite over time with laser irradiation
[Comment 5] There are some errors in the manuscript, please revise them carefully.
Reply: Thank you for pointing out our mistakes. The typo has been corrected as follows.
- nanopillars -> nanofillers---(Page number -2, line -28)
- 5W/cm2, 0.8-0.5cm2 -> 1.5 W/cm2, 0.8-0.5 cm2---(Page number -4, lines -14)
- 100-350cm-1, 1591cm-1 and 1347cm-1->100-350cm-1, 1591cm-1 and 1347cm-1---(Page number -5, lines- 10 to 12 and Page number -6, line-1)
- sp2, sp3->, sp2, sp3---(Page number -6, line- 30)
- 3200cm-1, 1730cm-1, 3500cm-1, 2925cm-1, 2845cm-1 ->3200cm-1, 1730cm-1, 3500cm-1, 2925cm-1, 2845cm-1---(Page number -7, lines- 6 to 9)
- ~10-11S/cm, the electrical conductivity changed from 4.72×10−8 to 1.07×10−6 and 4.66×10−6 S/cm -> ~10-11S/cm, the electrical conductivity changed from 4.72×10−8 to 1.07×10−6 and 4.66×10−6S/cm---(Page number -8, lines- 21 to 23)
- NIR laser (818 nm, power density 0.1 W/cm2) -> NIR laser (818 nm, power density 0.1 W/cm2) ---(Page number -9, line- 5)

Reviewer 3 Report
The authors systematically investigated mechanical, thermal, electrical properties and heat-generating performance of PU/CNT nanocomposite material. Overall, the work is detailed and the results from different aspects matched well to some extent. I’m sure certain efforts have been made by the authors and the presented data in the plots could give reference to the readers in the relevant field.
Most part is structurally organized, but in the discussion section, there is no comparison with other related CNT/PU composites, and some further analysis is missing about self-heating effect with the addition of CNT. Besides, some modifications listed below should be addressed. In the context of this manuscript, clearer and more precise writing will improve the paper.
P5L16 diffraction angle is 2θ,not θ.
P7P8 give comparison with properties of other CNT/PU composites
P8L12 add further analysis about self-heating effect with the addition of CNT
Author Response
Referee #3
The authors systematically investigated mechanical, thermal, electrical properties and heat-generating performance of PU/CNT nanocomposite material. Overall, the work is detailed and the results from different aspects matched well to some extent. I’m sure certain efforts have been made by the authors and the presented data in the plots could give reference to the readers in the relevant field.
Most part is structurally organized, but in the discussion section, there is no comparison with other related CNT/PU composites, and some further analysis is missing about self-heating effect with the addition of CNT. Besides, some modifications listed below should be addressed. In the context of this manuscript, clearer and more precise writing will improve the paper
[Comment 1] diffraction angle is 2θ,not θ
Reply: Thank you for this valuable comment. As per your suggestion we corrected the figure by writing 2θ in place of θ. The correction has been made in revised manuscripts.
Figure R5. XRD pattern of pristine CNT(P-SWCNT) and heat treated SWCNT(H-SWCNTs)
[Comment 2] give comparison with properties of other CNT/PU composites
Reply: Thank you for your valuable comment. The developed polyurethane composites with H-SWNTs showed excellent self-heating properties than the composites made by commercial SWNTs. According to previously reported data, the nanocomposites of commercially available SWCNT and polyurethane have the maximum self-heating temperature of 164°C, which has been increased up to 270°C by incorporating high crystalline SWCNTs even just in one minute. Also, for comparison, the data on PU composites using P-SW were added to the supplementary material.
[Comment 3] add further analysis about self-heating effect with the addition of CNT
Reply: Thank you for your kind comments. If the content of CNTs with developed surface structures is increased, it is expected that the exothermic properties will naturally be improved. However, when checking the exothermic results of P-SWNT (Fig. R4) and H-SWNT (Fig. 4), it can be seen that there is no significant difference in the maximum exothermic temperature at 3wt % and 5wt %. Therefore, in the case of this study, the filler filling threshold is judged to be around 5wt %. In order to induce improved exothermic properties, CNT modification in a different direction, such as hetero-element doping, seems to be necessary to increase the charge transfer. We have published research results on polyurethane composites with higher exothermic properties by doping CNT surfaces with boron.
Reference: Ha, Y.-M.; Kim, Y.-O.; Kim, Y.-N.; Kim, J.; Lee, J.-S.; Cho, J. W.; Endo, M.; Muramatsu, H.; Kim, Y. A.; Jung, Y. C., Rapidly self-heating shape memory polyurethane nanocomposite with boron-doped single-walled carbon nanotubes using near-infrared laser. Compos. B. Eng. 2019, 175, 107065.

Round 2
Reviewer 2 Report
The authors clearly responded to the comment.
I recommend accepting the paper in its present form.